# Position: Scaling LLM Agents Requires Asymptotic Analysis with LLM Primitives

Elliot Meyerson [1]   Xin Qiu [1]

## Abstract

Decomposing hard problems into subproblems often makes them easier and more efficient to solve. With large language models (LLMs) crossing critical reliability thresholds for a growing slate of capabilities, there is an increasing effort to decompose systems into sets of LLM-based agents, each of whom can be delegated sub-tasks. However, this decomposition (even when automated) is often intuitive, e.g., based on how a human might assign roles to members of a human team. How close are these role decompositions to optimal? This position paper argues that asymptotic analysis with LLM primitives is needed to reason about the efficiency of such decomposed systems, and that insights from such analysis will unlock opportunities for scaling them. By treating the LLM forward pass as the atomic unit of computational cost, one can separate out the (often opaque) inner workings of a particular LLM from the inherent efficiency of how a set of LLMs are orchestrated to solve hard problems. In other words, if we want to scale the deployment of LLMs to the limit, instead of anthropomorphizing LLMs, asymptotic analysis with LLM primitives should be used to reason about and develop more powerful decompositions of large problems into LLM agents.

## 1. Introduction

The turn to agents is well underway (Guo et al., 2024; Wang et al., 2024b). Now that large language models (LLMs) have crossed critical thresholds of general capability and reliability (Bubeck et al., 2023), it is natural to ask how they can be integrated into computational systems larger than themselves, in order to *do things*: "Stop speaking and act!" The term *agent* has quickly crystallized as the term

for LLM-based programs that carry out subprocesses within larger systems or act "autonomously" in virtual or physical environments. Unlike the general purpose LLMs from which they descend, LLM agents usually have a specific scope in which they are intended to operate, i.e., specific kinds of inputs they are expected to receive and corresponding kinds of outcomes they are expected to produce. Agent scopes vary in the precision with which they are specified, but the very act of specifying a scope, or *role*, for an agent allows it to be treated as a computational object that can serve as a component in the development of a larger system. The promise of LLM agents is not only that they have the capacity to affect change in the real world, but that by constructing increasingly larger systems of many agents, where each has its own complementary focus, we can dramatically expand the scale at which LLM-based AI is applied.

There has been a rush to start building such systems (Xi et al., 2025). In academia, LLM agents have been used in applications across the board, for example, in improving performance on standard benchmarks (Du et al., 2024), building teams for more effective software development (Qian et al., 2024; Wu et al.), simulating communities and human behavior (Park et al., 2023; Yan et al., 2024), and tackling the research process itself (Huang et al.; Lu et al., 2024; Yu et al., 2024). In industry, the impact (at least on strategy and priorities) has been arguably greater: while academics usually prefer minimal systems in which agent behavior can be most clearly understood, industry leaders find themselves faced daily with huge organizations with countless interconnected actors and components, and thus countless opportunities for agents to be applied (Hodjat, 2024; Urlana et al., 2024; Sypherd & Belle, 2024).

The core focus thus far in the development of such systems has been *"What is possible with the resources we have today?"*, not *"How efficient can it be in the limit of scale?"* Efficiency at scale can be understandably an afterthought when the goal is to build something that works and is impressive as quickly as possible, but **this paper argues that understanding the efficiency of LLM-agent-based systems at a fundamental algorithmic level is critical to achieving the scale such systems promise.** Specifically, missing from the study of such systems are *asymptotic analyses* that give

[1]Cognizant AI Lab, San Francisco, USA. Correspondence to: Elliot Meyerson <elliot.meyerson@cognizant.com>.

*Proceedings of the 42$^{nd}$ International Conference on Machine Learning*, Vancouver, Canada. PMLR 267, 2025. Copyright 2025 by the author(s).

a fundamental characterization of the *cost* (temporal, energy, or monetary) of running the system at *scale*, e.g., as the size, complexity, and number of problem instances increases to the limit. Although such analysis may seem esoteric when folks are just at the point of trying to get *something* to work, if such systems are really to scale to millions of users, millions of distinct task instances being solved, increasingly large and complicated tasks to be solved, and beyond, a formal understanding of the asymptotic behavior of these systems will be a key ingredient to guide research and development in the directions most likely to make such scale practically achievable. Put loosely, this is because solutions that are intuitive or off-the-shelf and appear to work well for initial problems being solved today may be inherently limited when considering the question of scale, and there may be alternative directions that require different kinds of research, directions which careful asymptotic analysis can illuminate. Asymptotic analysis has been critical to the scaling of systems and algorithms throughout the history of computation (Landau, 1909; Turing, 1936; Knuth, 1968; Aho & Hopcroft, 1974; Sedgewick, 1996; Montanaro, 2016; Cormen et al., 2022), and we see no reason why the scaling of LLM agents should be different.

**We argue that key to such analysis is the concept of the *LLM primitive***, i.e., treating the forward pass of an LLM as the atomic unit of computation we are interested in counting in our analysis. By abstracting away the internals of any particular LLM, the analysis can focus on the higher-level algorithmic behavior that emerges from the roles of agents within larger systems. Aside from clarifying the analysis, this abstraction ensures that any algorithmic efficiency improvements discovered at the agentic level are orthogonal to the ongoing efficiency improvements of LLMs themselves.

To flesh out the argument, this paper proceeds as follows: Section 2 sketches a basic framework for undertaking asymptotic analysis with LLM primitives (AALPs); Section 3 presents three example analyses using this framework, showing distinct cases where an asymptotic analysis leads to valuable insights that might otherwise go undiscovered; Section 4 engages with alternative views, i.e., ways AALPs might *not* in fact be required; Section 5 describes some of the critical research directions for developing AALPs into as useful a tool as possible; and Section 6 concludes.

## 2. Asymptotic Analysis with LLM Primitives

In order to make the argument of this paper concrete, this section sketches a minimal framework for asymptotic analysis with LLM primitives (**AALPs**). Although it elides many phenomena critical to the complete understanding of agentic LLM systems, this basic framework is sufficient to instantiate clear examples of the advantages of an asymptotic analysis-driven approach to system development, as

will be shown in Section 3. This framework can then serve as a seed for future research that develops a more complete approach, as discussed in Section 5.

The premise of asymptotic analysis is that by focusing on counting the executions of the fundamental operations whose cost we most care about, we can characterize the fundamental trends in cost as the system scales. Such analysis can reveal scaling insights that might be obscured by implementation details or cost overheads that dominate at the smaller scales that are empirically viable today.

Let us first define our central object of interest:

**Definition 2.1** (Language-based Algorithm). A language-based algorithm (or LLM-based Algorithm (Chen et al., 2024)), **LbA** for short, is any algorithm in which one or more of the computational steps are performed by an LLM.

In other words, an LbA consists of a set of LLM-based agents that work together to complete tasks.

In the classical asymptotic analysis of algorithms, the basic unit of computation is a *primitive*, an atomic operation whose executions can be counted characterize the behavior of the algorithm at scale. Standard choices of a primitive include addition or variable assignment, i.e., operations that can be performed in a single CPU cycle. This paper argues that the appropriate primitive for asymptotic analysis of agentic LLM-based systems is a single execution of an LLM, i.e., a forward pass that generates a single token $v \in V$. By treating LLMs as atomic, we can focus on the fundamental behavior of the LbA, orthogonal to ongoing improvements inside the LLMs themselves. We count only LLM operations because the computational resources to run LLMs are usually the most salient limiting factor in scaling LbAs. Since our focus is on scaling the deployment of LbAs, we can also ignore training cost, with the expectation that the cost of running inference with fixed models over a long period of time and massive scale will rapidly dominate the cost of training (Sardana et al., 2024) (a loosening of this assumption is discussed in Section 5.6).

For convenience, we assume the cost of an LLM execution depends linearly on the number of input tokens $n$ and the size of the model $m$. This assumption is based on the fact that industry APIs like those from OpenAI and Anthropic charge linearly per token (OpenAI, 2025; Anthropic, 2025). Although in principle the cost of a single forward pass through a transformer model scales quadratically with the input size (Vaswani, 2017), core optimizations like sparse attention (Tay et al., 2020) and representation caching (Luohe et al., 2024) have moved the practical cost closer to linear, as reflected in the API prices. If the cost is in fact superlinear, asymptotic separations between LbAs will generally be even sharper. So, for the purposes of this paper, we use the following definition:

**Definition 2.2** (LLM Primitive). An LLM primitive is an operation defined by an LLM $M$ of size $m$, such that a single application of $M$ to a string of length $n$ has cost $mn$.

Key to this definition is the idea that different LLMs $M_1, M_2, \ldots$ can have drastically different sizes $m_1, m_2, \ldots$ and thus drastically different costs. Notice that we use "cost" here in a general sense, it could refer to an estimate of the raw FLOPS, the economic cost (Chen et al., 2023; Shekhar et al., 2024), or the environmental cost (Bender et al., 2021; Faiz et al., 2024), and we expect all of these to be highly correlated. Why wouldn't one always use the smallest LLM available? Different LLMs have different sets of capabilities they can reliably perform. For an LLM primitive to be reliably applied in an LbA, the operation performed must be within the *capabilities set* of the LLM being used:

**Definition 2.3** (Capabilities Set). Every LLM $M$ has a corresponding capabilities set $C_M = \{c_1, c_2, \ldots\}$, which characterizes the scope of tasks $M$ can reliably perform.

In an LbA, different LLM-based operations might be performed by different LLMs with different capabilities. This separation of labor motivates the specialization of LLMs, i.e., one need not rely on a single giant AGI-like model to solve all tasks that can be specified in text, especially the most basic ones. A fully optimal LbA will use LLMs with minimal capabilities sets, since increasing the size of the set cannot decrease the minimal size of the required model.

Making the capabilities set explicit in the design of an algorithm has the auxiliary benefit of raising awareness of the trade-offs between two algorithms, i.e., if two algorithms have similar complexity, but one requires more accessible capabilities, that algorithm should be preferred. Note also that specifying a capability set is only necessary for the purpose of designing the abstract algorithm for complexity analysis; one need not enumerate all the capabilities of a real-world LLM, rather just verify that it has the required capabilities in a particular implementation. This idea is in line with the call for LLM *specifications* (Stoica et al., 2024).

The central tool for achieving asymptotic improvements in LbAs over a naïve application of LLMs is the expectation that large and complex jobs can be meaningfully decomposed, and that the impact of this decomposition will scale in a manner not accessible by the naïve approach. Sometimes the precise decomposition can be described analytically or deduced programmatically, but sometimes, especially when reasoning about abstract problems, it can be useful to simply assume we have a way of decomposing a problem, in order to make asymptotic analysis possible. Any meaningful benefits uncovered in the analysis then motivate the development of a practical implementation of the decomposition. We call this tool an *oracle decomposition*:

**Definition 2.4** (Oracle Decomposition). An oracle decom-

position is an assumed non-LLM operation that decomposes a problem in a specific way, without regard to how the operation is implemented.

Armed with this basic set of tools, one can proceed to comparatively analyze the asymptotic behavior of different LbAs on different problems, highlighting the kinds of places asymptotic improvements can be found.

Then, if LLMs themselves are developed in a way that alters their cost and/or capabilities, these results can be directly incorporated into AALPs to identify implications for particular LbAs. AALPs also provides a framework for analyzing LbAs that require capabilities that LLMs are not yet capable of, thereby allowing researchers to be more prepared for the future when it arrives.

Note that there has already been one promising first attempt at a more detailed and formal treatment of AALPs (Chen et al., 2024), and we refer the reader to that work for further example analyses, which complement those in Section 3. However, the goal of the present paper is not to develop a complete framework, but to present a clear argument with illustrative examples to bring the basic idea of AALPs to as broad an audience as possible. In other words, the present work is a call to action.

## 3. Example Asymptotic Separations

This section provides concrete examples of how AALPs can be applied to demonstrate massive asymptotic separations between LbA's when applied at scale. The goal of these examples is not to show challenging math, but simple results that immediately highlight the importance of such analysis. Our hope is that these simple examples will inspire future work that tackles more complex LLM-agent systems.

The three example problems in this section highlight different kinds of opportunities that would arise from having a solid complexity theory for LLM agents. The first highlights the criticality of understanding where to focus LLM size optimization in many-agent systems; the second highlights critical pitfalls of naïve anthropomorphic multi-agentization; and the third highlights issues that arise from applying LLMs to large and creative optimization tasks. See Table 1 for a high-level summary. Note that the analysis in this section is *intentionally loose*, and intended to be as rudimentary as possible, as the goal is simply to highlight the kinds of insights that can emerge, and argue that further development of AALPs will be critical in understanding how to scale real-world systems. To that end, we invite the reader to critique any assumption (implicit or explicit) made in the following analyses, with the hope that such critique will lead to more practical AALPs methodologies and thus more powerful LbAs.

| | $k$-Task (Sec. 3.1) | Debugging (Sec. 3.2) | Evolution (Sec. 3.3) |
|---|---|---|---|
| Agents | Generalist $M_g$
Delegator $M_d$
Specialists $M_s^1, \ldots M_s^k$ | QA Engineer $M_q$
Software Developer $M_d$ | Mutator $M_u$ |
| Optimistic | $y \leftarrow M_g(x)$ | **for** $i \in 0 \ldots b-1$ **do**
    bug $\leftarrow M_q(x)$
    $x \leftarrow M_d(\text{bug}, x)$ | **while** $f(x) < k$ **do**
    $x' \leftarrow M_u(x)$
    **if** $f(x') > f(x)$ **then**
        $x \leftarrow x'$ |
| Optimized | $i \leftarrow M_d(x_{0:c})$
$y \leftarrow M_s^i(x)$ | **for** $i \in 0 \ldots k-1$ **do**
    bug $\leftarrow M_q(x_{il:il+l})$
    **if** bug **then**
        $x_{il:il+l} \leftarrow M_d(\text{bug}, x_{il:il+l})$ | $i \leftarrow 0$
**while** $f(x) < k$ **do**
    $x' \leftarrow x$
    $x'_{il:il+l} \leftarrow M_u(x_{il:il+l})$
    **if** $f(x') > f(x)$ **then**
        $x \leftarrow x'$
        $i \leftarrow i+1$ |
| Improvement | $\Theta(k)$ | $\Theta(bk^2)$ | $\Theta(2^k k)$ |

*Table 1. Overview of Examples.* This table gives a high-level summary of the examples described in Section 3. It lists the LLM agents used, and gives pseudocode for the optimistic implementation (i.e., based on an intuitive belief in the power of LLMs) and the optimized one (based on a more careful LbA design (Def. 2.1)). The improvements result from applying AALPs to these implementations.

## 3.1. $k$-Task Routing

The section highlights potential scaling advantages of having specialized agents of asymptotically smaller size than highly-capable generalist agents. For this problem, suppose the input size is $n$ and the output size is $\Theta(1)$. Many common tasks like information retrieval, classification, and agentic single-action tasks fall under this specification. Suppose there are $k$ such distinct tasks of this form that we would like the system to be able to perform.

### 3.1.1. GENERALIST APPROACH

The simplest approach would be to use a single generalist many-task LLM agent $M_g$ of size $m_g$ capable of solving any of the tasks on its own. Processing the entire input and returning a constant sized answer will then cost

$$\Theta(m_g n). \tag{1}$$

### 3.1.2. DELEGATOR AND SPECIALISTS APPROACH

Suppose instead that we have a system consisting of a delegator $M_d$ of size $m_d$ and $k$ specialists $M_s^1, \ldots, M_s^k$, one for each of the $k$ distinct tasks. Several recent approaches take this kind of LLM-routing approach (Hu et al., 2024; Feng et al., 2025; Narayan et al., 2025). For simplicity, assume the specialists are all of equal size $m_s$. Suppose the delegator can determine which of the $k$ tasks the input string belongs to by observing only a constant number of metadata tokens prepended to the input, e.g., the delegator has been fine-tuned so that descriptions of the $k$ tasks need not be supplied in the prompt. Then, the cost of identifying

the task and solving it with the designated specialist is

$$\Theta(m_d) + \Theta(m_s n). \tag{2}$$

### 3.1.3. IMPLICATIONS

We can safely assume $m_d \leq m_g$, since solving all $k$ tasks almost certainly requires (implicitly or explicitly) identifying which task is being solved as a sub-capability.[1] So, as $n$ grows, the comparison of interest is $\Theta(m_g n)$ vs. $\Theta(m_s n)$, i.e., $m_g$ vs. $m_s$. From analysis of the neural scaling laws of LLMs (Kaplan et al., 2020), we can assume the required size of the LLM scales with the number of required capabilities. For example, let us assume that in the present setting, the required size of the LLM scales linearly with the number of tasks it can solve. Then,

$$\Theta(m_g) = \Theta(km_s) \implies \Theta(m_g n) = \Theta(km_s n). \tag{3}$$

That is, the speed-up from delegation and specialization scales linearly with the number of tasks. As LLM-agent-based systems become larger and larger, many hope that they will be able to tackle thousands of distinct tasks, if not more. So, the advantage of having many specialist agents becomes enormous.

Importantly, notice that this analysis shows that, for those developing the LLMs themselves, it should be much more

---

[1]Evidence for this can be found in mixture-of-experts models: In Mixtral 8x7B (Jiang et al., 2024a), each expert (analogous to a specialist) has $\approx$7B parameters, while the router (delegator) only has $\approx$32K, and the overall system outperforms a generalist LLM with $\approx$70B parameters; DeepSeek-V3 (Liu et al., 2024) routes across 256 experts, by simply comparing a single input token embedding to the "centroid" token embedding of each expert.

impactful to focus on minimizing the size of the specialist LLMs than minimizing the size of the delegator. This is because, even if the delegator is relatively huge, it is called so few times relative to the specialists, and the cost of any initial fine-tuning of the delegator is amortized over the lifetime of the system. In a real-world team of human developers, without such analysis, equal research and development resources might be allocated to both reducing the size of the delegator and reducing the size of specialists. For example, it could be very tempting to try to develop a relatively tiny classifier as the delegator, since it might seem like a relatively simple classification problem. Analysis like the above can preempt research on shrinking the delegator: It would be okay to take a giant off-the-self maximum-capability LLM as the delegator, fine-tune it once, and then focus on shrinking the specialists. This observation supports the more general idea that, in an optimal LLM *ecology*, we should expect orders of magnitude more executions of smaller models than of larger models (Nisioti et al., 2024).

### 3.2. Iterative Code Debugging

This section looks at another increasingly common and critical task for LLM-agents: code debugging. At a high level, the problem is that we have a large code-base with many bugs, and we would like to clean it up. For clarity of analysis, suppose the code is $n = kl$ tokens long, consisting of $k$ functions, the implementation of each being $l$ tokens long, and suppose there are $b$ bugs somewhere in the code, with no more than one bug per function. The goal is to provide an efficient LbA that fixes all the bugs.

#### 3.2.1. NAÏVE MULTI-AGENT APPROACH

Suppose we take an anthropomorphic multi-agent approach, akin to the kind of approach that has become currently quite popular for pushing the problem-solving performance of coding LLMs (Park et al., 2023; Qian et al., 2024; Wu et al.; Song et al., 2024). In such an approach, we instantiate multiple agents by prompting them with different roles, mapping each on to an employee in a human organization. For simplicity in the present analysis, we will use only two roles: quality assurance (QA) engineer $M_q$ and debugging specialist $M_d$ (we leave it as an exercise to the reader to extend/revise the analysis for other common roles found in multi-agent coding systems).

In this approach, we alternate between $M_q$ (of size $m_q$) identifying a bug and $M_d$ (of size $m_d$) fixing it. Suppose $M_q$ has the capacity to identify exactly one bug at a time, and $M_d$ has the capacity to fix exactly one identified bug. In the standard approach, the entire conversation history, including prior versions of the code are fed as input to each agent at each iteration. From an asymptotic perspective, such accumulation in the input may seem clearly inefficient, but

this is an approach taken in popular multi-agent frameworks today, such as autogen (Wu et al.). Suppose the QA engineer identifies bugs by producing constant size messages, while the debugging specialist rewrites the entire code every time it fixes a bug (as is also standard in current systems). Then, for the $i$th bug, the QA agent looks at all $i$ previous versions of the code and previous QA comments, yielding a total input length of $\Theta(in)$. So, the cost of QA over all $b$ bugs is

$$m_q(\Theta(n) + \Theta(2n) + \ldots + \Theta(bn)) = \Theta(m_q b^2 n). \quad (4)$$

Meanwhile, the debugging expert looks at $in$ tokens at each iteration and produces $n$ new tokens of code, yielding

$$m_d(in + (in + 1) + \ldots + (in + n)) = \Theta(m_d in^2) \quad (5)$$

at each iteration, and over all $b$ bugs:

$$\Theta(m_d n^2) + \ldots \Theta(m_d bn^2) = \Theta(m_d b^2 n^2). \quad (6)$$

So, the total cost of QA plus writing the debugged code is

$$\Theta(m_q b^2 n) + \Theta(m_d b^2 n^2) = $$
$$\Theta(m_q b^2 kl) + \Theta(m_d b^2 k^2 l^2). \quad (7)$$

We notice immediately that, due to the additional factor of $n$ for the debugging expert compared to QA engineer, similar to the implication highlighted in Section 3.1, it is much more important to focus on minimizing the size of the debugging expert than the QA engineer. It could be fine to use the biggest LLM available for QA, since its relative asymptotic cost is so small, especially when the code grows extremely large. However, the central implication from this section comes not from relative model size but from comparison to a more focused approach to the multi-agent problem decomposition, as is discussed next.

#### 3.2.2. FOCUSED MULTI-AGENT APPROACH

Let's suppose all functions in the code have completely correct, precise and constant size specification (e.g., in their docstring). This implies the full code can be fixed by fixing each function independently. Now, instead of having the QA agent look at the entire code every iteration, suppose it looks only at a single function. Then, it will overall look at the entire code only a single time, resulting in a cost of

$$\Theta(m_q kl). \quad (8)$$

Whenever a bug is identified in a function, the debugging expert then fixes the bug by looking at and rewriting only that single function, yielding a total bug-fixing cost of

$$\Theta(m_d l^2 b), \quad (9)$$

and thus the cost for the whole system is

$$\Theta(m_q kl) + \Theta(m_d l^2 b). \quad (10)$$

### 3.2.3. IMPLICATIONS

Since both the QA engineer and debugging expert in the focused version of the system have smaller jobs and less to keep track of, it would be reasonable to assume that they could be of a smaller size than their naïve counterparts. However, even supposing they are no smaller, we have relative speed-ups of $\Theta(b^2)$ for QA and $\Theta(bk^2)$ for the debugging expert. Since $n \geq b$, and the job of the debugging expert is intuitively more challenging than that of QA (since it has to actually produce correct code), the performance improvement that is likely to dominate the relative cost of the two approaches is that of the debugging expert:

$$\Theta(bk^2). \tag{11}$$

This is a huge speed-up, and highlights the power of breaking down solutions into the minimal possible bite-sized chunks that can be precisely specified for agents to operate on. This improvement becomes astronomical as the size of the code increases to the size of large industrial codebases. The apparent limitations of most existing multi-agent LLM approaches to relatively small codebases could be due in part to this scaling behavior, and thus pursuing the direction of extreme decomposition could be critical to scaling robust LLM-based coding systems.

This implication also generalizes to other applications where the goal is to iteratively refine large objects of interest, e.g., design or construction applications. The insights from a properly developed asymptotic analysis with LLM primitives could be critical to making such applications scale.

Notice that we made a strong and critical assumption of perfect code specification in the description of the focused multi-agent approach. The fact that this leads to such efficiency improvements is motivation to investigate whether LLMs are capable of generating such precise and correct specifications; if such focused precision is possible, it could have an enormous impact.

### 3.3. Evolutionary Optimization

Another area where LLM agents are increasingly being applied is in optimization, where the LLM is used as the engine of variation, i.e., given some existing solutions it is used to generate variations on these solutions that have a chance of being improvements with respect to some evaluation function (Lehman et al., 2023; Meyerson et al., 2024; Bradley et al., 2024; Yang et al., 2024; Romera-Paredes et al., 2024; Lee et al., 2025). Many of these applications have been developed under evolutionary optimization frameworks (Bäck et al., 1997; Wu et al., 2024). In such scenarios, there is some evaluation, or *fitness*, function $f(x)$ that we would like to maximize, and the LLM is responsible for generating solutions $x \in \mathcal{X}$. If $x$ is representable by text, then an LLM

can be applied for this role, and, in theory, any solution type is representable by text (Meyerson et al., 2024).

Let us suppose $f$ is of a special form that is common for analysis of evolutionary algorithms, i.e., it is a variant of the ONEMAX function (Doerr, 2020; Witt, 2013). Specifically, suppose each solution $x$ has length $n = kl$ tokens, such that it consists of $k$ blocks each of length $l$. Suppose each block is either "correct" or "incorrect", and the value of $f(x)$ is the number of correct blocks in $x$. Then, the maximum value of $f(x)$ is $k$, in the case that all blocks are correct. Note that in practice there may be many ways for a specific block to be considered correct.

### 3.3.1. GLOBAL MUTATION

Suppose we have an LLM-based mutation agent $M_u$ of size $m_u$, that, when applied to an input string $x$, alters every block of $x$ in *some* way, resulting in a 50% chance of that block now being "correct". Suppose we start with an arbitrary initial solution $x$, and run a greedy algorithm, where we iteratively apply the mutator to get a new solution $x'$, and replace $x'$ as our current solution if it has higher fitness $f(x') > f(x)$. Note, this is an instantiation of a $(1+\lambda)$-EA (Droste et al., 2002), the most often analyzed algorithm in the EA literature (Doerr & Neumann, 2021), which has been used to analyze the optimization of deep architectures in other settings (Meyerson & Miikkulainen, 2019; Meyerson et al., 2022).

Now, if $M_u$ is applied to the entire string $x$, the expected number of times it needs to be applied to generate a completely correct solution is $2^k$. Since each application of $M_u$ is to the whole solution, the generation of each new solution costs $\Theta(m_u n^2)$, and thus the total expected cost is

$$\Theta(m_u 2^k n^2) = \Theta(m_u 2^k k^2 l^2), \tag{12}$$

which is completely impractical even for any decently sized value of $k$. This huge expected cost could be a reason we have seen LLM-based optimization methods being applied mainly to small (though interesting and semantically complex) problems: They are usually applied to the entire solution all at once.

### 3.3.2. LOCAL MUTATION

Suppose instead we have a programmatic way of determining the boundaries of blocks (oracle decomposition), and, instead of applying $M_u$ to the entire solution at once, it is applied to each block individually and in sequence, i.e., generating one updated block at a time. Then, each application will cost $\Theta(m_u l^2)$, and the total expected cost will be

$$\Theta(m_u k l^2). \tag{13}$$

### 3.3.3. IMPLICATIONS

The relative performance improvement of the local mutation approach over the global one is enormous:

$$\Theta(2^k k). \tag{14}$$

Notice also that this improvement is before any consideration of model size, i.e., before considering that a capable local mutator agent might be substantially smaller than a global one. Of course, local mutations will not be sufficient for all evaluation functions, particularly those that are non-convex. However, they can still be quite powerful and practical tools, and gradient descent itself is in the most basic sense a local variation operator.

Again, we invite the reader to look back at the assumptions made in this section and see where there is room for improvement. For example, we assume that solutions here can be broken down into components programmatically, but in practice, one may need to assume something fuzzier and more approximate.

In any case, this example points again to the criticality of problem decomposition in a dramatic case where it leads to super-exponential improvements at scale. This simple analysis shines a light on the potential of a specific subfield (i.e., optimization driven by LLMs) to benefit from careful asymptotic analysis of their constituent LbAs.

Finally, notice that, in contrast to the prior two problems, this section considered a case where the LLM agent acts reliably, but not deterministically, hinting at further opportunities to generalize this kind of analysis.

### 3.4. Exercises

The above three examples are intended as representatives of a vast space of possible LbAs for which asymptotic analysis is critical. To become more deeply acquainted with the position of this paper, we invite the reader to conduct their own analysis on variants of these problems as well as other problems, such as the below:

- Sorting sets of large documents, e.g., for organizing legal arguments or prioritizing resumes (especially long and plentiful resumes of AI agents themselves).

- Solving a text-based puzzle, e.g., reordering the shuffled sentences of a novel to their original order.

- Writing and refining a paper for submission to a conference dedicated to AI-produced AI research.

- Creating a policy proposal, e.g., to fight climate change or for pandemic response.

- Designing and booking an optimal $k$-night vacation given a set of constraints.

We are curious to see what themes emerge from researchers with differing backgrounds approaching asymptotic analysis with LLM primitives in different ways on these problems and others. Further examples of problems ripe for such analysis can be found in recent work (Chen et al., 2024).

## 4. Alternative Views

Now that we have made the case for the adoption of asymptotic analysis with LLM primitives in the understanding and development of scaled LLM-based agentic systems, it is worth reflecting on possible counterarguments to this position. Such an engagement can yield the identification of strengths and weaknesses of the position, and further promising directions of thinking thereby. We invite the reader to come up with alternative views beyond the ones discussed here, as there are surely more and stronger ones.

**Alternative View 1:** *All the implications of such analysis are things people would do anyway.*

The intuition behind this view is that the algorithmic properties of agentic LLM systems are not that complicated, and optimal cost optimization of such systems will follow naturally from the kinds of refinements researchers and engineers tend to do anyway. If true, this would be a huge blow to the argument for AALPs, since it would not add any practical value to LLM-based systems, it would just serve as a source of esoteric exercises for the algorithmically inclined. However, we believe the evidence provided in Section 3 is sufficient to dispel this view, i.e., insights from the analysis will have real-world impact on where human and computational resources are focused, and thus have an impact on the timeline of deployment for scaled LLM agents.

**Alternative View 2:** *Direct optimization for cost is good enough, without thinking about asymptotic behavior.*

The view here is that for any given real-world LLM agentic system, the general aymptotic limiting case is not as important as the simple reality of optimizing for the problems the particular instantiation of the system faces today. This view is similar to Alternative View 1, except that it's focused on particular instantiations of a system. For example, we might imagine an automated external optimization process that refines the design of an agentic system, e.g., modifying agent scopes, modifying the communication channels between agents, and swapping out the underlying LLMs of different agents. However, although such a process could yield meaningful speed-ups in special cases, the innovations discovered in such a system would likely not generalize to greater problem scales, i.e., in the scaling limit. This failure of generalization is common in systems overly-optimized to overly-specific use cases (Tan & Le, 2019). Although it can be possible to mine general insights from such specialized

innovations, most of the optimizations should be expected to lead to dead-ends at scale. AALPs is the tool to free cost optimization from such tempting dead-ends.

**Alternative View 3:** *We don't need LLM primitives: Existing cost measures are enough—such as simply counting tokens, GPU hours, or floating point operations.*

This view raises the question of what level of granularity is most useful for understanding LbA behavior. If we count tokens without regard to model size, we miss out on the increasing differences in model scale required for different capabilities; if we count GPU hours, total activated LLM parameters, or floating point operations, the results will be conditional on the particular hardware and LLM internal implementations available at the time. By encapsulating model cost and capabilities, AALPs provides the right level of abstraction for reasoning about the costs of scaled agentic systems, orthogonal to the ongoing development of LLMs.

**Alternative View 4:** *We don't need this for superintelligence, and once we get superintelligence, it will optimize everything better than humans ever could.*

Prominent AI researchers are preparing for the possibility of super-intelligence within a few years (Grace et al., 2024; Altman, 2024; Hendrycks et al., 2025; Fortson, 2025). We believe this presents the strongest case against AALPs. The view here is that progress is being made so fast at the core of LLMs that we are bound to hit a level of general intelligence and automated AI research capabilities beyond that of human AI researchers before we need to scale modular and compute-optimized many-agent systems in the real world. Once such superintelligence is achieved, the superintelligent AI itself will be capable of performing asymptotic analysis as it sees fit, and will be otherwise responsible for developing future algorithmic insights leading to scaled agentic systems. As there is some evidence of slowing progress at the core of LLM development (Cyran, 2024; Booth, 2024), and there is always uncertainty of whether a particular research direction will hit a fundamental wall, we believe that the orthogonal AALPs approach, which foregrounds the advantages of many-agentness, is a worthwhile complement to the direct approach of LLM improvement. We also believe that since AALPs is such a fundamental tool for understanding the behavior of agentic LLM-based systems, and agentic decomposition is fundamentally essential to the optimal use of resources in the limit (i.e., a self-optimized superintelligence will not use its full power to do single-digit addition), such a superintelligence must rely on AALPs as it decides how to optimize its usage of computational resources in the physical universe, and, for those human researchers developing and applying AALPs methods, having a superintelligence use your conceptual technology is not such a bad academic legacy.

## 5. Research Directions

The goal of this paper is to serve as a catalyst for the adoption and development of asymptotic analysis with LLM primitives into the research and development of LLM-based agentic systems. As a result, the discussion has focused on high-level principles and minimal motivating examples to highlight the central advantages of such analysis as clearly as possible, and many critical considerations have been thus far ignored. This section enumerates several such considerations, to serve as a scaffolding for future research.

### 5.1. Identifying Useful Assumptions

As alluded to throughout this paper, AALPs provides a template for systematically reasoning about the efficiency of LLM-based agentic systems, but any specific application of AALPs depends on the particular assumptions adopted, and which assumptions are most useful for driving progress is an open question. For example, are the linear scaling assumptions in Definition 2.2 and Section 3.1.3 reasonable? Is it reasonable to disregard relatively large constants like the system prompt and explanations surrounding an answer, or assume they can be distilled away? Is it reasonable to assume functional independence of solution components as is done in Sections 3.2 and 3.3, or should we explicitly deal with approximations in cases of messy problem decompositions or in the absence of an oracle? Moving forward, it will be important for AALPs to handle more nuanced assumptions based on complexities arising in practice.

### 5.2. Stochasticity and Error-correction

For simplicity, the analysis in Sections 2 and 3 assumed that the constituent LLMs can execute capabilities in their scope reliably, i.e., the scenario where the LLM does not successfully fulfill its role is not considered. Of course, LLMs today do make errors, even surprisingly simple ones (Basmov et al., 2023; Williams & Huckle, 2024; Lehman et al., 2025). A more complete analytical framework will take the likelihood of such errors into account. Initial work has developed core ideas in how error probabilities can compound even in the roll-outs of single LLMs (Dziri et al., 2024); further work is needed to extend such insights to the realm of LLM agents. For example, ideas of error correction, e.g., from information theory (Hamming, 1950; Pless, 2011), could be integrated into agentic systems to reduce the impact of errors at scale, akin to the centrality of error-correction in quantum computing (Lidar & Brun, 2013; Roffe, 2019). One open problem key to making such error correction work is developing LLM agents whose errors are decorrelated. Established theory of randomized algorithms will be an essential resource for this project (Motwani & Raghavan, 1996; Mitzenmacher & Upfal, 2017).

## 5.3. Asynchrony and Distributed Algorithms

Similarly, the examples in Section 3 did not consider potential (temporal) cost benefits from agents running asynchronously, in a distributed manner, or otherwise in parallel. Such parallelization is extremely natural, especially if the central mechanism for achieving asymptotic improvements is problem decomposition. Similar to the case of randomized algorithms above, existing methodologies for asymptotic analysis in parallel, asynchronous, and distributed systems (Akl, 1989; Baudet, 1978; Santoro, 2006), as well as classical multi-agent systems (Ferber & Weiss, 1999; Van der Hoek & Wooldridge, 2008; Hodjat et al., 1998), will be an invaluable resource, and the incorporation of such aspects will be required to clarify the full scope of the advantages of carefully designed LLM-agent systems.

## 5.4. Automatic Decomposition

A central motivating theme throughout this paper is the expectation that the jobs of large and complex LLM-based agentic systems can be usefully decomposed into subproblems, and some such decompositions lead to asymptotic improvements over others. In the examples in Section 3 we assumed that effective decompositions were available a priori or could be deduced programmatically (i.e., without the use of LLMs). In practice, especially in more open-ended systems where the full spectrum of tasks the system might need to solve is not known beforehand, having automatic decomposition methods will be critical to maintaining/maximizing asymptotic performance. Such methods will likely rely on LLMs themselves, in which case their cost must be incorporated into AALPs. There has been some initial work on automatic agent decomposition (Wu et al.; Song et al., 2024), but most is based on intuitive zero-shot approaches; much more asymptotically advantageous approaches should be possible.

## 5.5. Extensions to Other Modalities

This paper argues that AALPs is required to scale LLM agents, but as agentic systems naturally grow alongside AI models that incorporate a greater and greater range of modalities (Team et al., 2023; Liang et al., 2024), the capabilities afforded by these modalities will naturally be incorporated into agentic systems; this process has already begun (Jiang et al., 2024b; Gao et al., 2024; Sarch et al., 2024). AALPs does not immediately extend to other modalities, due to its basis in LLM primitives, but similar techniques could be used to encapsulate the usage of each modality as a primitive with an assigned cost. A simple implication could be that many visual tasks (when performed at scale) should be executed by compact vision-only models instead of full-fledged multi-modal foundation models.

## 5.6. Online Learning and Adaptation

This paper has focused on the case where all LLM training happens beforehand, and thus can be viewed as a fixed cost in comparison to the inference (forward-pass) costs of running an LLM-agent system at scale over a long timeframe. However, for more adaptive systems, it may be essential to allow some form of agent learning during deployment. Such learning could consist of the collection and refinement of memories (represented in text) as the system encounters new scenarios (Park et al., 2023; Wang et al., 2024a), or fine-tuning updates through gradient descent (Hu et al., 2023; Rannen-Triki et al., 2024; Ding et al., 2023; Han et al., 2024). In any case, the computational cost of such learning, dependent on the size of the constituent LLM, will need to be carefully incorporated into AALPs.

## 5.7. Ethics and Sentience

Finally, one side benefit of highly-decomposed modular many-agent LLM systems is a potential massive reduction in future machine suffering. As centralized AI systems become larger and more capable, there are compelling arguments that at some threshold of capabilities *sentience has to emerge*, at which point the AI system will likely endure astronomical levels of suffering (Metzinger, 2021; Saad & Bradley, 2022; Chiang, 2021). One avenue to minimizing the likelihood of such suffering is to build highly-decomposed many-agent systems that achieve the same positive impact of a centralized system, but, by minimizing the capabilities and scope of each AI, reducing the chance that sentience will emerge. This argument is related to a position paper from last year, arguing that reducing the scope of AI memory will reduce the chance of suffering, since memory is such a critical ingredient for suffering in humans (Tkachenko, 2024). If complemented by further research in how sentience can emerge, AALPs could thus play a role in reducing the chance of immeasurable machine suffering.

## 6. Conclusion

This paper has claimed that asymptotic analysis with LLM primitives will be critical to the project of scaling LLM-based agentic systems. The argument for this position was developed through contextualization with existing work, concrete examples of how such analysis can lead to impactful insights, and engagement with alternative views. Several important directions were then highlighted as a guide for future research. Our hope is that this discussion will serve as a catalyst, motivating developers of LLM agents to carefully consider how their systems might scale, establishing an exciting research project within the cutting edge of AI for algorithmists and other academics who do not have access to hyperscale computing resources, and, most of all, to accelerate the real-world positive impact of LLM agents.

## Acknowledgments

We would like to thank the Cognizant AI Lab research team for useful discussions in the development of this work, and, in particular, Dan Fink and Giuseppe Paolo for early inspiring discussions before this work started to take form.

## Impact Statement

This paper presents work whose goal is to advance the field of artificial intelligence. There are many societal consequences that potentially follow from such work. One main advantage of the approach advocated in this paper is the potential reduction in future machine suffering, as discussed in Section 5.7. Another advantage, arising from the fact that AALPs leads naturally to maximally decomposed problem-solving, is that decomposed systems can be much more interpretable, auditable, and therefore safer than a single opaque centralized AI carrying out the same task, a point that has been acknowledged in prior work (Khot et al., 2021; Sharkey et al., 2025).

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
