# OpenReview forum: "Position: Scaling LLM Agents Requires Asymptotic Analysis with LLM Primitives"
_ICML.cc/2025/Position_Paper_Track — ICML 2025 Position Paper Track poster_

### Official Review · Reviewer_3BhU · 2025-02-16

**Significance:** 2
**Argument Clarity:** 2
**Rating:** 2
**Confidence:** 2

**Questions:**

- Could such asymptotic analysis be employed for other types of networks that would be much less costly than LLMs?

**Discussion Potential:**

2

**Paper Summary:**

This paper advocates the adoption and development of asymptotic analysis with LLM primitives into the research and development of LLM-based agent systems.
The authors outline concrete examples of how such asymptotic analysis can be impactful (k-Task routing, iterative code debugging, greedy optimization and exercices), and provides alternative views.
Several directions are highlighted for future research.

## update after rebuttal

The authors discussed about super-intelligence in their rebuttal, and I have not found the conclusions to be convincing. My score was "weak reject" before the rebuttal and I am now definitely convinced about keeping this score.

**Position:**

Yes

**Position In Title:**

Yes

**Related Work:**

3

**Strengths And Weaknesses:**

Strengths:

- the advocated position is supported by concrete examples explaining why such asymptotic analysis could be impactful
- it is likely to inspire discussion as LLMs are widely used in ML
- the paper cites related works in an appropriate way

Weaknesses:

- the technicalities related to asymptotic analysis and briefly outlined but might deserve more explanation for the novice reader
- the alternative view about how super-intelligent machines could perform the asymptotic analysis seems slightly irrelevant at the first glance, since LLMs are nowadays unable to perform very simple mathematical tasks such as numbers comparison or multiplication

**Support:**

2

---

> ### Author Rebuttal · Authors · 2025-04-01
>
> *Response to suggestion that the technicalities of asymptotic analysis could be elaborated upon for the novice reader*
>
> Thanks for the suggestion. We will provide more explanation in Section 2 around the paragraph discussing classical primitives vs. LLM primitives, and cite some introductory references there. At a high-level, the premise of asymptotic analysis is that by focusing on counting the executions of the fundamental operations whose cost we most care about, we can characterize the fundamental trends in cost as the system scales. Such analysis can reveal scaling insights that might be obscured by implementation details or cost overheads that dominate at the smaller scales that are empirically viable today.
>
> *Response to comment that alternative view discussing super-intelligence could be irrelevant*
>
> We agree that super-intelligence can appear far away when looking at current models. However, our position is advocating for work in a specific direction over a period of years; recent surveys show that a significant portion of AI researchers believe that we are relatively close to achieving super-intelligence [1] and the most prominent frontier AI organizations are preparing for the possibility of super-intelligence within a few years [2, 3, 4], so we believe it is useful to consider the views of these readers. In the revised version of the paper, we will explicitly add this motivation in Alternative View 3.
>
> [1] K. Grace, et al. 2024 “Thousands of AI Authors on the Future of AI”. arXiv:2401.02843 [cs.CY]
>
> [2] S. Altman. 2024 "The Intelligence Age." https://ia.samaltman.com/
>
> [3] D. Hendrycks, E. Schmidt, and A. Wang. 2025 "Superintelligence Strategy: Expert Version." arXiv:2503.05628 [cs.CY]
>
> [4] D. Fortson. 2025 “Anthropic chief: ‘By next year, AI could be smarter than all humans’”. The Times.
>
> *Response to question on whether asymptotic analysis could be employed for networks much smaller than LLMs*
>
> Yes! Asymptotic analysis can be applied to any algorithmic system that can be theoretically scaled. This paper focuses exclusively on LLM-based systems because we believe understanding how their costs will scale is a critical concern (practical, economic, and environmental) that deserves focused attention. As touched on in Section 5.4, it would be interesting to see how the method extends to other models.

---

> > ### Comment · Reviewer_3BhU · 2025-04-02
> >
> > I would like to thank the authors for their detailed reply. I will maintain my score.

---

> > > ### Author Response · Authors · 2025-04-02
> > >
> > > Thanks for the reply. If you agree we have sufficiently addressed the weaknesses you noted, would you consider increasing your score to align with the other reviewers? Or do you have any additional questions about our rebuttal? Thanks!

---

### Official Review · Reviewer_j73p · 2025-02-18

**Significance:** 3
**Argument Clarity:** 4
**Rating:** 4
**Confidence:** 3

**Questions:**

na

**Discussion Potential:**

3

**Paper Summary:**

The paper argues that scaling large language model (LLM) agents requires asymptotic analysis with LLM primitives to optimize efficiency. It proposes treating the LLM forward pass as the atomic unit of computational cost and using this approach to analyze and improve the decomposition of complex problems into subproblems for LLM agents. The authors provide examples demonstrating how asymptotic analysis can reveal significant efficiency improvements and suggest that this approach is crucial for scaling LLM-based systems.

## update after rebuttal
No update. According to the author's response, they will make improvements in the revised version.

**Position:**

Yes

**Position In Title:**

Yes

**Related Work:**

4

**Strengths And Weaknesses:**

S:

The paper introduces a novel framework for analyzing the efficiency of LLM-based systems using asymptotic analysis with LLM primitives.

The paper provides several concrete examples (e.g., k-task routing, iterative code debugging, and greedy optimization) that illustrate how asymptotic analysis can lead to substantial efficiency improvements. These examples highlight the potential benefits of the proposed approach.

The paper suggests that highly decomposed LLM agent systems could reduce the risk of machine suffering by minimizing the capabilities and scope of each AI agent. This aligns with broader ethical concerns in AI development.

W:

The examples provided in the paper rely on several assumptions, such as linear cost scaling and perfect problem decomposition. These assumptions may not hold in real-world scenarios, potentially limiting the applicability of the insights gained from the analysis.

**Support:**

3

---

> ### Author Rebuttal · Authors · 2025-04-01
>
> Thank you for seeing the overall value of our work and understanding that identifying practical assumptions is a key requirement to making AALPs an effective tool. In the revised version of the paper, we will add a new subsection in Section 5 to make it even clearer that determining which kinds of assumptions are most useful is a critical research direction.

---

### Official Review · Reviewer_EuV9 · 2025-03-14

**Significance:** 3
**Argument Clarity:** 3
**Rating:** 4
**Confidence:** 4

**Questions:**

1. Could you elaborate on practical strategies for achieving or approximating oracle decompositions in real-world scenarios where perfect decompositions are challenging? The problem might be significant when the process is heuristic or dependent on LLM behaviors, alike cases in Section 3.3.
2. The proposed "LLM primitives" are not a novel concept. Could you discuss how your approach differs from existing methods that statistically calculate the number of prompt & completion tokens on a specific dataset?
3. Asymptotic analysis is a powerful tool but only when the properties of the system are large enough. How would the analysis handle the case when some constants are not relatively small, e.g., the system prompt, or are not controllable, e.g., LLM generated content for even simplest tasks, to be ignored?

**Discussion Potential:**

3

**Paper Summary:**

This paper argues that scaling LLM-based agentic systems effectively requires an asymptotic analysis approach that considers LLM forward passes as primitives. The core position is that current intuitive or anthropomorphic decompositions into LLM-based agents are suboptimal, and proper asymptotic analysis with clearly defined LLM primitives can reveal significant efficiency improvements. Through illustrative examples such as task delegation, code debugging, and greedy optimization, the authors demonstrate substantial potential for computational efficiency gains when problem decompositions are guided by rigorous asymptotic analysis. The idea and claims in the paper is interesting and well-argued. Though with some limitations, the paper provides a valuable perspective on the future of LLM-based agentic systems.

## update after rebuttal

I maintain my scores given the authors' responses are aligned with my expectation. And I believe the position this paper holds has an impact in the related domain. However, I insist that more related works about multi-agent systems, system costs measurement, and etc. need to be discussed in addition to the current version of the manuscript. Therefore, I keep the scores and not change for higher ones.

**Position:**

Yes

**Position In Title:**

Yes

**Related Work:**

1

**Strengths And Weaknesses:**

### Strengths

- The paper clearly states a well-defined position that is highly relevant to current trends in scaling LLM-based systems.
- The argument is strongly supported by explicit illustrative examples, clearly demonstrating the potential efficiency gains from applying asymptotic analysis.
- The discussion acknowledges alternative views explicitly and thoughtfully engages with potential criticisms.
- Clearly identified directions for future research provide a valuable roadmap for further exploration.

### Weaknesses

- While the examples provided are illustrative, they rely heavily on idealized assumptions (such as oracle decompositions or perfect specifications). The practical feasibility is neglected, e.g., common practices in LLM inference system designs including KV caching, paralleling, MoE, etc., efficient techniques for LLMs including quantization, pruning, and uncommon practices beyond single-token prediction, e.g., multi-token prediction, non-autoregressive decoding, etc.
- The practice of using LLM primitives for asymptotic analysis is valuable, but the paper does not provide a clear connection between this approach and the scalability of LLM-based agent systems beyond scaling the number of inference times. The LLM-based agent system could benefit from scaling other aspects, e.g., model size, post-training, inference-time optimization, or even scaling the number of agents without significantly increasing the number of inference tokens.
- The exercise part may not be necessary in the main body. It could be replaced by a more detailed discussion on the limitations of current complexity analysis approaches in LLM-based agentic systems.
- Instead of talking about the non-requiring positions as alternative views, the paper should focus more on the other possible pathways to realize the aymptotic analysis, e.g., with prompt&completion token counting, activated parameters, float number computation, GPU hours, etc. Current alternative views are weakly supported and not well-argued.
- The related work review is not comprehensive enough. The paper should include more recent works on LLM-based multi-agent systems associating to examples in Section 3, and provide examples of how some of these works could benefit from the proposed asymptotic analysis framework.
- Although claiming broad applicability, the paper primarily considers text-based scenarios. The generalizability to multimodal or non-text-based tasks requires further elaboration.

However, the paper is well-written and provides a valuable perspective on the future of LLM-based agentic systems, and the weaknesses are not that significant in my perspective.

**Support:**

3

---

> ### Author Rebuttal · Authors · 2025-04-01
>
> Thanks for the feedback! Responses to comments/questions:
>
> *Examples rely on idealized assumptions*
>
> Yes, this paper intentionally adopts simplified assumptions in order to spark further discussion of what the most useful kinds of assumptions would be. The revised version of the paper will include a new subsection in Section 5 to make it even clearer that determining which kinds of assumptions are most useful is a critical research direction.
>
> *Connections to other kinds of LLM scalability*
>
> Yes, as noted in the Introduction, the approach in this paper is orthogonal to other avenues of improving LLM efficiency. We will expand on this complementarity in the revised version. It arises from how AALPs encapsulates model cost and capabilities: If models are scaled in a way that alters their cost and/or capabilities, e.g., increasing cost leading to new capabilities or decreasing cost without losing capabilities, these results can be directly incorporated into AALPs analysis to identify implications for particular LbAs. AALPs also provides a framework for analyzing LbAs that require capabilities that LLMs are not yet capable of, thereby allowing researchers to be more prepared for the future when it arrives.
>
> *Including other ways of measuring LLM system cost as an Alternative View*
>
> Great idea! We will add an Alternative View to cover this. At a high level, it is a question of what level of granularity is most useful for reasoning about long-run costs. If we count tokens without regard to model size, we miss out on the increasing differences in model scale required for different capabilities; if we count GPU hours, the results will be conditional on the particular hardware available at the time.
>
> *References to more recent LLM-based agentic systems in Examples*
>
> Sections 3.2 and 3.3 include references to LLM-based coding and optimization systems. In Section 3.1, we will add references to recent work on LLM routing, including GraphRouter [1], Minions [2], and RouterBench [3]. Feel free to suggest other missing references that would be valuable to add. Thanks!
>
> [1]  T. Feng, Y. Shen, and J. You. 2025 “GraphRouter: A graph-based router for LLM selections.” ICLR 2025. \
> [2]  A. Narayan, et al. 2025 “Minions: Cost-efficient Collaboration Between On-device and Cloud Language Models”. arXiv:2502.15964 [cs.LG] \
> [3] Q. J. Hu, et al. 2024 “Routerbench: A benchmark for multi-llm routing system.” arXiv:2403.12031 [cs.LG]
>
> *Generalizability to multimodal or non-text-based tasks requires elaboration*
>
> Yes, this paper focuses exclusively on text-based tasks. In the revised version we will make it clearer in Section 5.4 that the approach will not transfer directly to other modalities, and that the easiest modalities to extend to will likely be those in which data is encoded as tokens.
>
> *Practical strategies for decomposition*
>
> Right, in practice, decomposition will often rely on heuristic techniques, either programmatic or LLM-based. In either case, if many input instances have the same structure (as can be the case in large-scale enterprise applications), the decomposition can be reused across input instances, so the initial cost of decomposition can be amortized. Of course, in practice, these decompositions will only be approximations of the oracle, so an important research direction for AALPs is developing methods for handling approximations; a discussion of this need will be added to Section 5.
>
> *How the approach differs from methods that count tokens on a specific dataset*
>
> The main difference between AALPs and methods that statistically calculate the number of prompt & completion tokens on a specific dataset is that we are interested in abstracting away empirical details in order to characterize the behavior of systems at scales beyond those we have so far been able to test empirically. The hope is that such asymptotic analysis will reveal scaling insights that might be obscured by implementation details or relatively large overheads at smaller scales. We will make this clear in the revised version.
>
> *Handling cases where some constants are not relatively small*
>
> This is a great question. If the system prompt is so large that it has a substantial impact on cost, even when scaling to large and complex problems, then perhaps it should not be treated as a constant after all, e.g., if it is required to grow in order for the LLM to handle larger or more complex instances. We will mention this in the aforementioned new subsection of Section 5 on determining the most useful assumptions.
>
> This observation also suggests that if the system prompt or extraneous generated output are dominating the overall cost, fine-tuning the model to avoid the need for these additional tokens would make it possible to realize the benefits of asymptotic scaling properties on relatively smaller instances. This reinforces one of the central takeaways of the paper: that smaller specialized models are critical to scaling LLM-based agentic systems.

---

### Official Review · Reviewer_ZxoU · 2025-03-15

**Significance:** 4
**Argument Clarity:** 2
**Rating:** 4
**Confidence:** 4

**Questions:**

See weaknesses

**Discussion Potential:**

3

**Paper Summary:**

The paper argues that scaling LLM-based agents efficiently requires rigorous asymptotic analysis using LLM primitives. It critiques the current intuitive approach to decomposing complex tasks into simpler sub-tasks assigned to different LLM-based agents, suggesting that such decompositions are often anthropomorphic and not necessarily optimal. By proposing to treat the LLM forward pass as the fundamental computational unit, the authors advocate for a theoretical framework to analyze the computational efficiency of multi-agent LLM systems systematically. Through illustrative examples, they demonstrate significant asymptotic improvements achievable by carefully analyzing and optimizing task decomposition. The paper concludes by highlighting essential directions for future research, including error handling, distributed computing, automatic decomposition, multimodality, and ethical considerations, emphasizing that understanding the theoretical limits and scaling behaviour is crucial for deploying LLM agents effectively at scale.

**Position:**

Yes

**Position In Title:**

Yes

**Related Work:**

4

**Strengths And Weaknesses:**

Strengths:
- Novelty in the topic and approaches: This paper covers an important topic of LLM Agent efficiency and introduces the innovative concepts of LLM primitives from asymptotic analysis. This may help further research in LLM scaling.
- Well-illustrated examples: the inclusion of detailed and easy-to-follow illustrative examples clearly demonstrates the value and potential of their asymptotic analysis.
- Comprehensive discussion of research directions: The authors further discussed the future directions. Giving practical suggestions is appreciated.

Weaknesses:
-  Flaws in the assumptions in examples.
** In Section 3.1, I believe in many practical uses, $M_d$ is very much similar to $M_g$ because task routing is often a very difficult tasks. And the complexity of such tasks should $\Theta(m_d n)$ instead $\Theta(m_d)$ because it often needs to see all the tasks. Based on such assumptions, reducing the size of delegator is still meaningful. Furthermore, in this case, to what extent $m_d$ is smaller than $m_g$ is crucial. This means, how complex the decomposed tasks are is important. And in many cases, people use a powerful LLM because the tasks are hard to decompose.
 ** In Section 3.2, I think one important assumption here is functions are independent and therefore you can debugging them independently. But in many real cases, they are not.
- (Minor) Presentation of examples can be better: The authors could use figures or flowcharts to illustrate examples instead of text and equations.

**Support:**

3

---

> ### Author Rebuttal · Authors · 2025-04-01
>
> *Response to comments on the assumptions in Sections 3.1 and 3.2*
>
> First, thanks so much for engaging with and interrogating the example assumptions; this is exactly the kind of discussion we wish to spark with this paper, as noted in the second paragraph of Section 3. In the revised version of the paper, we will add a new subsection in Section 5 to make it even clearer that determining which kinds of assumptions are most useful is a critical research direction.
>
> With regards to the comment that the delegator and generalist might need to be of similar size, we note that, in the $k$-task routing problem considered in Section 3.1, the delegator only needs to classify the input as one of $k$ known tasks, which is generally much easier than open-endedly decomposing an input into subtasks.
>
> Evidence for this can be found in mixture-of-experts models. For example, in the seminal Mixtral 8x7B model [1], each expert (analogous to a specialist) has ~7B parameters, while the router (delegator) only has ~32K, and the overall system outperforms a generalist LLM with ~70B parameters. Similarly, DeepSeek-V3 [2] routes across 256 experts, by simply comparing a single input token embedding to the learned “centroid” token embedding of each expert.
>
> We agree that for an off-the-shelf LLM delegator, it must see $\Theta(k)$ tokens to be aware of which $k$ tasks it can route to (recall that $n$ is the length of the input sample, and $k$ is the number of different tasks). However, awareness of the tasks could also be fine-tuned into the delegator, and this one-time cost of fine-tuning would be amortized over the lifetime of the system. We will add a discussion of this to the paper.
>
> Regarding the assumption in Section 3.2 about independence of functions:
>
> Yes! This is a very important assumption, thanks for engaging with the work to identify underlying assumptions! We will mention this assumption more explicitly in the paper, i.e., that it assumes the functions all have correct specifications and that their correct implementation will result in a correct implementation of the full program, i.e., so they can be debugged independently. This is an extreme case, whereas in practice programs will fall on the spectrum between independence and complete inter-dependence. We also want to reemphasize that the practice of identifying and refining such assumptions is a key part of effective asymptotic analysis, so we are happy that our work has inspired the identification of these assumptions!
>
> [1] Albert Q. Jiang et al. 2024. “Mixtral of Experts”. arXiv:2401.04088 [cs.LG]
>
> [2] DeepSeek-AI 2024. “DeepSeek-V3 Technical Report”. arXiv:2412.19437 [cs.LG]
>
> *Response to idea of adding figures or flowcharts to illustrate examples*
>
> Thanks for the feedback! With the extra page allowed in the final version, we will have space to include algorithmic diagrams for each of the examples.

---

### Decision · Program_Chairs · 2025-04-30

**Decision:**

Accept (poster)

**Comment:**

This paper advocates for using rigorous asymptotic analysis – instead of relying on intuition – when designing complex LLM-based systems that issue multiple calls to diverse models.

All the reviewers agree that the position is supported by several good examples, and that the paper is likely to spark a discussion.

The weakness is that the analysis relies on strong assumptions that may not hold in real world, but this is somewhat understandable for an early take on a new theoretical framework. Other weaknesses mentioned by the reviewers are minor.

Three reviewers find the paper interesting, one – not.